# Deep Learning for Classifying Physical Activities from Accelerometer Data

**DOI:** 10.3390/s21165564

**Published:** 2021-08-18

**Authors:** Vimala Nunavath, Sahand Johansen, Tommy Sandtorv Johannessen, Lei Jiao, Bjørge Herman Hansen, Sveinung Berntsen, Morten Goodwin

**Affiliations:** 1Department of Science and Industry Systems, University of South-Eastern Norway, Hasbergsvei 36, Krona, 3616 Kongsberg, Norway; 2CAIR, Department of ICT, University of Agder, Jon Lilletunsvei 9, 4879 Grimstad, Norway; Sahand.johansen@gmail.com (S.J.); tommy.s.johannessen@gmail.com (T.S.J.); lei.jiao@uia.no (L.J.); morten.goodwin@uia.no (M.G.); 3Department of Sport Science and Physical Education, University of Agder, Universitetsveien 25, 4630 Kristiansand, Norway; bjorge.h.hansen@uia.no (B.H.H.); sveinung.berntsen@uia.no (S.B.)

**Keywords:** classification, deep learning, health, machine learning, accelerometer data, sensors, physical activity, feed-forward neural network, DNN, recurrent neural network, RNN, UCI

## Abstract

Physical inactivity increases the risk of many adverse health conditions, including the world’s major non-communicable diseases, such as coronary heart disease, type 2 diabetes, and breast and colon cancers, shortening life expectancy. There are minimal medical care and personal trainers’ methods to monitor a patient’s actual physical activity types. To improve activity monitoring, we propose an artificial-intelligence-based approach to classify physical movement activity patterns. In more detail, we employ two deep learning (DL) methods, namely a deep feed-forward neural network (DNN) and a deep recurrent neural network (RNN) for this purpose. We evaluate the two models on two physical movement datasets collected from several volunteers who carried tri-axial accelerometer sensors. The first dataset is from the UCI machine learning repository, which contains 14 different activities-of-daily-life (ADL) and is collected from 16 volunteers who carried a single wrist-worn tri-axial accelerometer. The second dataset includes ten other ADLs and is gathered from eight volunteers who placed the sensors on their hips. Our experiment results show that the RNN model provides accurate performance compared to the state-of-the-art methods in classifying the fundamental movement patterns with an overall accuracy of 84.89% and an overall F1-score of 82.56%. The results indicate that our method provides the medical doctors and trainers a promising way to track and understand a patient’s physical activities precisely for better treatment.

## 1. Introduction

Physical activity (PA) is defined as “any bodily movement produced by skeletal muscles that result in energy expenditure above resting level” [1]. Physical activity recognition has been required by several real-life applications, such as monitoring older people, lifelong systems for monitoring energy expenditure and supporting weight-loss programs, and digital assistants for weightlifting exercises, etc. [2]. It is well known that physical inactivity is a risk factor for a variety of chronic diseases, particularly diabetes, cardiovascular disease, obesity, and depression [3,4,5,6]. There are minimal methods for medical care and personal trainers to monitor a patient’s actual physical activity types and training diaries where they commonly use logs. Today’s de facto method is personal diaries, of which the accuracy and credibility can be put into question, as these could be either intentionally or unintentionally subject to social desirability and recall bias [3]. Therefore, a reliable and objective movement registration method is vital.

In recent years, much research has been done to recognize the PA based on inertial sensor data from body-worn and smartphone sensors accelerometer, and gyroscope [2]. This paper focuses on the use of body-worn tri-axial accelerometers to collect activities and movement patterns where the raw data is converted into activity count variables, which is further used to classify physical activity intensity and energy expenditure [3].

There is a myraid of human activity recognition algorithms in the literature [7,8,9,10,11] many of which can recognize complex activities from wearable sensors. Still, the accuracy and precision of the algorithms depend on the number of activities, activity type, and data collection available. Static activities (e.g., sitting, standing) can be easily be separated from dynamic activities (e.g., running, jogging). However, many algorithms struggle with high precision, especially when there is overlap in the feature space, such as the separation between climbing stairs and descending stairs, and the separation between brushing teeth and combing hair. This is challenging because of the high similarity in the feature space and their related action pattern [7].

This paper explores the use of dense and recurrent deep learning techniques for activity recognition by using two datasets. The first dataset is the publicly available dataset which contains labeled accelerometer data recordings acquired from UCI Machine learning Repository [12]. It is a dataset for activities-of-daily-life (ADL) collected through wrist-worn accelerometers. The dataset has five categories and 13 daily living activities. The second dataset is collected ourselves from eight voluntary participants wearing hip-worn accelerometers, which contains labeled accelerometer data recordings of ten movement patterns of ADL, where some of the movement patterns are different variations of a movement. In contrast to most literature, our approach copes with activities with overlapping feature space, i.e., similar activities, and yields an overall higher precision than comparable algorithms. A system with precision is desirable for both high-performance systems and general monitoring systems in general since accurately detecting particular activities is of utmost importance. This is especially true in rehabilitation when whether or not a patient does particular activities and for how long (e.g., brushing your teeth, walking up stairs, walking down stairs) is highly relevant [10,11,13,14].

The rest of the paper is organized as follows. Section 2 provides the existing literature in the field of human activity recognition from the body-worn accelerometer data. The considered research methodology, including data acquisition process, use of different classifiers’ network architecture, and the descriptions of the publicly available dataset and the acquired dataset are presented in Section 3. Section 4 presents and discusses the experimental results obtained using two datasets. Finally, a conclusion and future research developments are given in Section 5.

## 2. Related Work

In the literature many researchers have applied deep learning in health care for classification [15,16,17,18,19], prediction [20,21] and diagnosis [22]. If we consider the literature on classification, recognizing, and predicting physical activities from body-worn accelerometer data, many attempts have been made over the past years. So, this section presents the existing literature on the use of deep learning for classifying physical activity using wearable sensor-based accelerometer data recordings from both wrist-worn and hip-worn accelerometers along with UCI HAR dataset which was acquired from UCI Machine learning repository.

### 2.1. UCI Har Dataset

Ronald et al. [23] proposed a deep learning model called iSPLInception which was based on Inception-ResNet architecture from Google to classify 6 activities of UCI HAR dataset. The activities include three static postures (standing, sitting, lying), and three dynamic activities (walking, walking downstairs and walking upstairs). Through experiments their proposed model achieved the highest test accuracy of 95.09% and F1 score of 95% performance. However, not all the 14 specific activities in UCI dataset are classified.

Steven Eyobu et al. [24], used a deep long term short term memory (LSTM) neural network architecture to check how feature representations and the augmentations on activity recognition accuracy can influence. To check that, the authors proposed a feature extraction approach combined with the data augmentation ensemble and used UCI HAR dataset to classify three activities (standing, sitting and walking). Their experimental results showed that the proposed extraction approach provided the best performance improvement in accuracy of 52.77%.

Chen et al. [25], proposed a feature fusion framework to combine handcrafted features with automatically learned features by a deep algorithm for HAR. Then, taking the regular dynamics of human behavior into consideration, they developed a maximum full a posteriori algorithm (MFPA) to further enhance the performance of HAR. Their experimental results showed that the proposed approach achieved 96.44% overall accuracy for 6 activities (Walking, upstairs, downstairs, sitting, standing, and laying) of UCI HAR dataset and 98.67% on 6 activities (Walking, fast walking, walking upstairs, walking down stairs, running, static) of their self-collected data set.

Shaohua Wan et al. [26], designed a deep learning architecture based on a smartphone inertial accelerometer for HAR. In this architecture, whenever the participants perform typical daily activities, a smartphone collects the sensory data sequence, extracts the high-efficiency features from the original data, and then obtains the user’s physical behavior data through multiple three-axis accelerometers. In addition, a real-time human activity classification method based on a convolutional neural network (CNN) was proposed, which uses a CNN for local feature extraction. Finally, CNN, LSTM, BLSTM, MLP and SVM models were applied on 6 activities (Walking, walking upstairs, walking down stairs, sitting, standing, and laying ) of the UCI HAR dataset. The experimental results showed that 1D CNN, 2D CNN, 2 layer LSTM, 3-layer LSTM, Bidirectional LSTM, 1D CNN+ LSTM achieved the overall accuracy of 65%, 76%, 85%, 91%, 93%, 77%, 76%, respectively.

Bruno et al. [27] proposed a framework for the recognition of motion primitives by using Gaussian Mixture Modeling and Gaussian Mixture Regression models. These models were used to classify 7 activities such as climbing the stairs, drinking from a glass, getting up from the bed, pouring water in a glass, sitting down on a chair, standing up from a chair, and walking. The experimental results showed the classification accuracy for climbing the stairs was 93.34%, drinking from a glass was 83.34%, getting up from the bed was 66.67%, pouring water in a glass was 80%, sitting down on a chair was 93.34%, standing up from a chair was 83.34%, and walking was 70% with using dynamic time warping and mahalanobis distance. Whereas, with only mahalanobis distance, the classification accuracy for climbing the stairs was 85.78%, drinking from a glass was 86.59%, eating with fork and knife was 96.67%, getting up from the bed was 61.64%, pouring water in a glass was 71.55%, sitting down on a chair was 100%, standing up from a chair was 93.96% and walking was 79.91%.

Nilay Tufek et al. [28], developed an activity recognition system by using accelerometer and gyroscope data. Deep learning methods such as Convolutional Neural Network(CNN), Long-Short Term Memory (LSTM), few classical machine learning algorithms and their combinations were implemented in this work. The experimental results show that the on the UCI HAR dataset, with using 3 layer LSTM model, 97.4% accuracy rate was achieved for the overall classification of 7 activities (Walking, jogging, lying, standing, falling, upstairs, down-stairs).

Murad et al. [29], proposed LSTM-based deep RNNs (DRNNs) to build human activity recognition models for classifying activities and developed architectures based on deep layers of unidirectional and bidirectional RNNs. These models were then tested on the benchmark datasets UCI-HAD, USC-HAD, Opportunity, Daphne FOG, Skoda to validate their performance and generalizability for a large range of activity recognition tasks. The study results on the UCI-HAD dataset showed that the unidirectional DRNN model achieved the 96.7% classification accuracy, CNN got 95.2%, SVM got 96.0%, sequential extreme learning machine got 93.3%.

Hassan et al. [30], the researchers’ goal was to accurately and automatically recognize and classify physical activity through wearable sensors by utilizing a Deep Belief Network (DBN) model. The authors used a publicly available dataset from the UCI machine learning repository for the experiments. For the comparison purpose, they used the SVM model. The experimental results showed that the DBN model achieved an overall accuracy of 97.5% with 40 hidden units for 12 activities (standing, sitting, lying, walking, walking downstairs and walking upstairs, stand to sit, sit to stand, sit to lie, lie to sit, stand to lie, lie to stand ). At the same time, SVM achieved an accuracy of 94.12%.

### 2.2. Self-Collected Datasets from Wrist-Worn and Hip-Worn Accelerometers

Trost et al. [31], implemented logistic regression model for predicting physical activities of wrist-worn and hip-worn accelerometer data. The collected data was from 52 children and adolescents who completed 12 activity trials that were categorized into 7 activity classes: Lying down, sitting, standing, walking, running, basketball, and dancing. In their study, features were extracted from 10-s windows and fed to the model. The study results showed the classification accuracy for the hip and wrist was 91.0% ± 3.1% and 88.4% ± 3.0%, respectively.

Mannini et al. [32], worked on classifying physical activities such as sitting, walking, lying, stair climbing, standing, running, cycling using on-body accelerometer data using single-frame classification algorithms: Naïve Bayesian (NB), SVM, Binary decision tree (C4.5), Gaussian Mixture Model (GMM), Logistic classifier, Nearest mean (NM), k-NN, parzen classifier, and ANN (multi-layer perceptron). The study results showed the classification accuracy for the NB model was 97.4%, GMM model was 92.2%, Logistic model was 94.0%, Parzen model was 92.7%, SVM model was 97.8%, NM model was 98.5%, k-NN model was 98.3%, ANN model was93.0%, C4.5 model was 93.0%.

Allen et al. [33] investigated two classification methods based on Gaussian mixture model (GMM) and a heuristic rule-based method to identify standing, sitting, lying, and walking using a triaxial accelerometer attached to the waist. The experimental results showed that GMM system was found to achieve a mean accuracy of 91.3%, distinguishing between three postures (sitting, standing and lying) and five movements (sit-to stand, stand-to-sit, lie-to-stand, stand-to-lie and walking), compared to 71.1% achieved by the Heuristic system. In this work, heuristic rule-based methods had problems in identifying sitting and standing as well as the transition between the two positions, GMM method also had problems in identifying the standing.

Sani et al. [34], presented an analysis of deep and shallow feature representations for accelerometer data on human activity recognition (HAR) with data collected from the wrist and thigh with activities such as walking, jogging, up Stairs, Down Stairs, Standing, and Sitting. They considered the three types of representations hand-crafted, frequency transform, and deep features, including two-hybrid approaches for HAR, and used CNN and introduced CNN hybrid approaches, i.e., CNN-SVM CNN-kNN. The classification performance of the models was measured using the F1-score. The results obtained show that the classification performance with data collected from the wrist got the highest F1-scores at 0.850 with CNN-SVM and 0.845 with CNN-kNN, respectively, and 0.839 with CNN.

Unlike the above works, our work in this paper classifies PAs using two different deep learning algorithms, i.e., DNN and RNN. In addition, when compared to the results mentioned above, we divided the raw data into three other sliding windows, such as three, five, and ten second windows. Further, in [29], the researchers used the RNN algorithm for the classification. However, the datasets they used are different from the datasets we used in this paper. Moreover, not all the authors classified all the 14 specific activities of the UCI HAR dataset. Further, the feature extraction methods are different from what we have used in our work.

## 3. Materials and Methods

In this section, we present the considered methodology including data collection and the used classifiers’ network architecture. Figure 1 summarizes the different steps of our adopted approach.

### 3.1. Data Collection

In this study, the classification of movement patterns is done on two datasets. The first is a publicly available dataset which contains labelled accelerometer data recordings acquired from UCI Machine learning Repository [12]. It is a dataset for activities-of-daily-life (ADL) collected through wrist-worn accelerometers including 5 categories and 13 daily living activities namely brushing teeth, combing hair, climbing stairs, descend stairs, walking, drinking from glass, pouring water into glass, eating with knife and fork, eating with spoon, using telephone, getting out of the bed, lying down in the bed, standing up from chair, sitting down on chair. This data set is here by referred to as the wrist-worn dataset.

The second dataset is collected ourselves through hip-worn accelerometers called ActiGraph, model GT3X+ from eight voluntary participants which contains labelled accelerometer data recordings of ten movement patterns of free-living ADL, where some of the movement patterns are different variations of a movement, such as different speeds of walking. The collected movement patterns are cycling, jogging, laying still, sitting, sitting in a vehicle, sitting relaxed, walking stairs, standing, walking fast, and walking normal. Before using the accelerometers, the participant were given instructions to perform activities in their own way without specific constraints. The second dataset is here by referred to as the hip-worn dataset.

### 3.2. Data Pre-Processing

In this study, we used the sliding window technique to extract features from raw data. The raw data is divided into three types of sliding windows of three-, five-, and ten second windows, respectively. More details on the sliding window process is described in Section 3.2.1 and Section 3.2.2.

#### 3.2.1. Wrist-Worn Dataset

The wrist-worn dataset consists of 14 different motion primitives which were performed by 16 volunteers. To test and examine the performance of the used algorithms, these motion primitives are divided into broader, less complex, ADL categories. The correlation between these ADL categories and the motion primitives are shown in Table 1.

The distribution of the data-points for these ADL’s and motion primitives are shown in Figure 2 and Figure 3, where the ADL distribution is on the left and the motion primitives on the right. When we looked at the ADL distribution, it can be seen that it is not evenly distributed as three categories; mobility (1), feeding (2) and functional transfers (4) which consists of above 100,000 data-points each. While personal hygiene (0) and communication (3) only consists of about 50,000 and 20,000 data-points, respectively. The overall distribution of the movement primitives can be considered somewhat even, with the exceptions of walking, eating with spoon, and lying down, as most of them are within the range of 20,000 to 40,000 data-points.

As mentioned above, the raw data is divided into sliding windows, i.e., a three second, a five second- and a ten second- sliding window in order to be used for training the algorithms. Whenever the accelerometers gather the data, it collects 32 data points each second.

In the sliding windows, the three second sliding window differs from the other two sliding windows, as the shift for each window also is set to three seconds. It is created by collecting three seconds of data (which is a total of 96 data-points) then sliding three seconds to create the next sliding window. Thus, there is no overlapping of data points within this type of sliding window. Dividing the dataset into these three second sliding windows yields a dataset consisting of approximately 4000 sliding windows distributed.When generating the five second sliding windows, the shift for each window is set to one second. Thus, each sliding window contains an overlap of four seconds to the previous window. This yields a dataset of approximately 10,000 sliding windows.

The final type of sliding window is ten seconds of data, also sliding by one second for each window. This creates a dataset consisting of approximately 6000 sliding windows.

#### 3.2.2. Hip-Worn Dataset

The hip-worn dataset consists of ten movement patterns performed by eight volunteers, where some of the movement patterns are different variations of a movement including different speeds of walking. The collected movement patterns are cycling, jogging, laying still, sitting, sitting in a vehicle, sitting relaxed, walking stairs, standing, walking fast, and walking normal. This dataset consists of 1.6 million data-points distributed among the categories as shown in Figure 4.

Similar to the wrist-worn dataset, the hip-worn dataset is also divided into three types of sliding windows, i.e., three-, five-, and ten-second windows, respectively. The hip-worn dataset is unevenly distributed and most of the data-points are in the relaxing categories lying down, and sitting. This pattern in the distribution is carried over to the sliding window datasets. Generating the dataset for three second sliding windows is done same as the wrist-worn, three seconds of data shifting is done with three seconds. However, the accelerometers used on this dataset collects 30 data-points each second, giving the three second sliding window dataset around 15,000 sliding windows with the distribution.

The dataset is created with five- and ten- second sliding windows with a shift of one second, the same as for the wrist-worn dataset. Both types of sliding window creates a dataset with approximately 54,000 sliding windows. After the data pre-processing, the data is fed to the deep learning algorithms. The description of the used deep learning networks architecture can be seen in the below Section 3.3.

### 3.3. Deep Learning Networks Architecture

In this paper, two deep learning algorithms DNNs and RNNs are used for classifying the activity movement types. The network structure of both used DNNs and RNNs is described below.

The first used network structure is the deep feed-forward network which is shown in Figure 5. The model consists of five layers i.e, an input and output layer, and three hidden layers. The three hidden layers have a constant number of cells, more specifically, the first hidden layer has 512 cells, the second hidden layer has 258 cells, and the last hidden layer has 128 cells. While the input and the output layer adapt to the dataset. The input layer always fit the length of the samples, while the output layer always have *n* cells, where *n* is the number of classes that need to be classified.

During the training phase, the input data is split up into smaller batches of 100 elements. The model predicts a label for each element and the weights and biases are adjusted from the sum of the errors between the predicted labels and true labels. The testing dataset is not split up into smaller batches. Thus, the entire testing dataset is used for calculating the performance metrics.

The second considered network structure is the recurrent neural network which consists of two components, the recurrent element of the network and a fully connected network. The first component, the recurrent element, consists of cells which in our case are Gated Recurrent Unit’s (GRU). Each of these cells are connected to each other and they all have a fixed number of states. The number of cells, and the number of states each cell has is provided as a parameter to the network. The recurrent element of the network returns two values the current state of the recurrent network $S_{t}$, and the output of each state $O_{t}$. The output of each state be an array of length *t*, where *t* is the number of items in the time-series that is sent into the network.

The last element of the outputs is used and sent forward to the fully connected network. The fully connected network is much smaller than the previous model and only consists of two layers. The input layer, which takes the input from the recurrent network and an output layer that is used to predict the class. A simplified illustration of the recurrent neural network model is shown in Figure 6.

## 4. Results

The results of our used models are presented and discussed in this section. The results are compared to other state-of-the-art methods. In this work, our experiments were conducted on Intel i7 7700K @ 4.2 GHz CPU, Nvidia GeForce GTX 1070 GB GPU, and Corsair 16 GB GB @ 3000 MHz RAM and several types of experiments were done by using four different learning rates, i.e., 0.1, 0.01, 0.001 and 0.0001. These learning rates are tested in combination with three well known optimizers, i.e., Adagrad, Adam and Stochastic Gradient Descent (SGD). Furthermore, these combinations of optimizers and learning rates are tested for all three types of sliding windows, i.e., three, five and ten seconds, both for the basic and specific movement categories. Considering the number of data-points available in the datasets, all experiments are stopped after 2000 training steps. Thus, reducing the chance of over-fitting through repeatedly training on the same data.

### 4.1. Deep Artificial Neural Network

Throughout this section, a detailed explanation of the best performing results with regards to the overall F1-score is given. For each type of sliding window (three-, five- and ten- seconds), the best performing combination of optimizer and learning rate be given for both the basic and specific movement categories.

Low learning rates are shown to be more suited for movement recognition, for both the Adagrad- and the Adam-optimizer. When using a high learning rate of 0.1, both the Adagrad and Adam optimizer get a low F1-score. The recall of these results shows that the algorithm guesses “randomly”. The probable cause of this might be the high learning rate. Thus, the algorithm diverges from the global minimum, which prevents it from learning the patterns of the movements. Furthermore, the result of the SGD optimizer is to some extent unexpected. However, the result can be explained through the used structure of the DNN. The model uses Rectified Linear Unit (ReLU) as its activation function. Using ReLU with SGD often leads to vanishing gradients, which stagnates the learning of the algorithm.

#### 4.1.1. Experiments with Wrist-Worn Dataset—Basic Movement

**Three seconds sliding window:** Testing the different hyper-parameter combinations on the three second sliding window shows the highest achieved overall accuracy and F1-score are 80.94% and 78.46%, respectively. This result is reached when using a low learning rate 0.0001 with the Adam optimizer.

Figure 7b shows a confusion matrix for the results of each movement type. This matrix is created to evaluate the algorithm, and to interpret why the algorithm achieved its results and where the incorrect classifications occurs. The recall (R), precision (P), and F1-score (F1) percentages about the classification of the movement types with 3-s window are shown in Table 2.

The classification inaccuracies for three second sliding windows are understandable considering the data distribution. The most distinct pattern in the matrix is that each category is to some extent inaccurately recognized as a functional transfer, even without it being the category with the largest amount of data-points. However, the probable cause of this pattern is the fact that the data is collected through a wrist-worn accelerometer. Each of the 14 participants might have different personal traits when moving, especially hand movements. This could affect the classification. Thus, giving F-transfer a lower precision score and can be seen in Table 2.

The ADL category which is inaccurately classified the most is the communication category, which contains the least amount of data-points. This is in line with what we expect, given the nature of the activities and placement of the sensor. Furthermore, when it is incorrectly classified, it is often interpreted as similar movement patterns, either hygiene or feeding. F-transfer is the category with the second highest percentage of inaccurate classification. Again, some of these classification might be affected by traits of the participants, thus classifying it as feeding. On the other hand, F-transfer are often classified as mobility, which is a more comparable category with regards to the movement types in those categories.

**Five second sliding window**: Using the Adam optimizer combined with a low learning rate of 0.0001, on the five second sliding window, the DNN achieves an overall accuracy of 86.01% and an overall F1-score of 82.37% and can be seen in Table 5. There are clear patterns in the inaccurate classifications of the ADL categories shown in the confusion matrix in Figure 7a. The recall (R), precision (P), and F1-score (F1) percentages about the classification of the movement types with 5-s window are shown in Table 2.

The DNN model incorrectly classifies the communication data as feeding (20% of the time), similar to the three second sliding window. Considering the resemblance in the movement patterns when using a telephone and eating, this confusion is understandable, especially when seeing the difference in the data distribution.

Explaining the miss-classification of the functional transfer category is partially based on assumptions. The two most commonly classified classes are mobility and feeding. As mentioned for the three second sliding window, the specific movement types in the mobility class and the functional transfer class (see Table 1) are comparable. It is understandable that getting up from a chair or the bed is interpreted as one of the mobility movements. However, looking at possible reasons for the DNN to interpret functional transfers as feeding, the most reasonable explanation would, as mentioned, be the placement of the accelerometer.

**Ten seconds sliding window:** The Adam optimizer and a low learning rate of 0.0001 is again the best performing combination when testing it on ten second sliding windows. Achieving an impressive overall accuracy of 92.4% and an overall F1-score of 89.43% (see Table 5). Looking at the confusion matrix in Figure 7c, it is clear to see that the ten-second sliding window allows the DNN model to observe and distinguish the differences between these basic movement patterns. The probable cause of the lower F1-score of communication is its low number of data-points which influences its precision score. Thus, predicting 5% of the hygiene data-points as communication heavily affects the precision score. The recall (R), precision (P), and F1-score (F1) percentages about the classification of the movement types with 10-s window are shown in Table 2.

#### 4.1.2. Experiments with Wrist-Worn Dataset—Specific Movement

As the number of movement types are increased in the specific movement distribution of the UCI dataset, the complexity of recognizing them also increases. An important point given that we ultimately want to predict all types of free-living (or at least the predominant types) activities with acceptable precision. In addition, the distribution of data-points is lower for each categories as they are no longer combined as for the basic distribution.

**Three seconds sliding window:** The best result achieved for three second sliding windows for specific movement is an accuracy of 59.93% and a F1-score of 56.59% (see Table 5). This result is achieved by the Adagrad optimizer with a learning rate of 0.01. Analyzing the confusion matrix in Figure 8a, the most difficult movement to recognize is lying down in bed. The recall (R), precision (P) and F1-score (F1) percentages about the classification of the specific movement types with 3-s window are shown in Table 2. Considering that it is the movement with the second lowest amount of sliding windows. Thus, it is to some extent expected.

The other reasonable miss-classified movement patterns are full body movements. Both climbing and descending stairs are often wrongly interpreted as walking, which is understandable as walking is the largest category in terms of data-points and the movements are comparable, which again is not surprising since, from a bio-mechanical view, both sensor signals are very similar. In addition, the two functional transfers, sitting down and standing up, are also interpreted as walking. Furthermore, descending stairs is either wrongly recognized as either walking or descending stairs. The accelerometer registers the g-forces, and as it is placed on the wrist. Thus, the patterns on climbing and descending stairs are relatively equal as the axes changes when rotating the hand.

Another conspicuous miss-interpretation is when the algorithm recognizes eating with spoon as pouring water into a glass. The assumption here is that the collected data of pouring water is from a mug. Thus, the resemblance in hand movements affects the algorithm. As an example, when eating soup one slowly moves the spoon from the plate, up towards the mouth before one tilts the spoon inside the mouth. The same pattern goes for pouring water from a mug into a glass. First the mug is lifted, then tilted to pour the water.

**Five seconds sliding window:** Testing the DNN on the five second sliding window, the highest performing hyper-parameter combination is the Adagrad optimizer and a learning rate of 0.01. Achieving an overall accuracy of 67.83% and an overall F1-score of 62.11% (see Table 5). The results of the five second sliding window are similar to the results of the three second sliding window, where most of the movement categories have a slight increase in their recall score (see Table 2). As mentioned in the Section 3.2.1, each five second sliding window consists of some overlapping data. Allowing the algorithm to more easily recognize the patterns of the movements. However, the movements which were miss-interpreted as walking are more frequently interpreted incorrect. This, pattern has a correlation to the distribution, as these movement categories, climbing and descending stairs, have fewer sliding windows for five seconds than for three seconds. Thus, the algorithm has fewer samples to train on, which affects the result and this can be seen in the confusion matrix results in Figure 8b.

**Ten seconds sliding window:** For ten second sliding window, Adagrad combined with a learning rate of 0.01 gives an overall accuracy of 81.29% and an F1-score of 66.75% (see Table 5). The recall (R), precision (P) and F1-score (F1) percentages about the classification of the specific movement types with 10-s sliding window are shown in Table 2.

Looking at the results shown in the confusion matrix in Figure 8c, it follows the same patterns as the five second sliding window. The recall score averagely increases due to the increase of the window size. As the window size increase, the amount of sliding widows decrease for the climbing- and descending- stairs movements. Thus, further decreasing their recall score, as they are more consistently interpreted as walking. The most distinguish pattern in the ten second sliding window confusion matrix is the miss-interpretation of standing up. It is approximately 40% of the time interpreted as getting out of bed. Indicating that the similarity of getting out of the bed, and standing up from a chair is quiet high when monitoring ten seconds of those movements.

### 4.2. Recurrent Neural Network

In this section, we discuss the best results of the Recurrent Neural Network (RNN) experiments. As for the DNN, a detailed explanation of the best combination of hyper-parameters is given for each sliding window, for both the basic and specific movement categories. In addition, the results for the secondary dataset, and the hip-worn data are explained.

The hyper-parameters used for the RNN are same as the DNN. However, two additional parameters are tested, i.e., number of cells in the RNN, and the size of the cells. The RNN is run with four and eight cells for each of the learning rates. Furthermore, these were both tested with two different sizes for the cells, i.e., 16 and 32 respectively.

Looking at the result overviews, lower learning rates often perform better than higher. However, Adagrad performs surprisingly bad with the lowest learning rate. The assumption here is that the number of training steps are to low for the Adagrad optimizer to learn the patterns with such a low learning rate. The same pattern for the SGD optimizer are shown for the RNN; SGD does not learn with the ReLU cells, except it is able to achieve relatively impressive results with the lowest learning rate.

#### 4.2.1. Experiments with Wrist-Worn Dataset—Basic Movement

The best performing results with regards to the overall F1-score for the basic movements for each sliding window type are explained below.

**Three seconds sliding window:** Running a RNN, consisting eight cells of size 32, with the Adam optimizer and a low learning rate of 0.001, an overall accuracy of 84.89% and an overall F1-score of 82.56% is achieved. When these results are compared to the DNN results (see Table 2, there is an increase of approximately 4%.

Looking at the confusion matrix in Figure 9a, the results again show that whenever a sliding window is miss-interpreted, it is often guessed as a hand movement. As an example, the category with the lowest amount of sliding windows, i.e., communication, is often guessed as feeding. This is understandable because of the similarity in movements, and also due to the fact that feeding is the category with the second most sliding windows.

The other noticeable result is the interpretation of functional transfers (see Table 3). As discussed in previous sections, the placement of the accelerometer can be used as an explanation. Placing it at the wrist is probably affecting the movement pattern, as a small hand gesture can have an influence on the algorithms. Thus, the assumption is that functional transfers is classified as feeding due to the accelerometer placement. The recall (R), precision (P), and F1-score (F1) percentages about the classification of the basic movement types using RNN for 3-s sliding window are shown in Table 3.

**Five seconds sliding window:** Using a learning rate of 0.01 and the Adagrad optimizer with the five second sliding window achieves an impressive 94.65% overall accuracy and a 93.03% overall F1-score which is tabulated in Table 5. The recall (R), precision (P) and F1-score (F1) percentages about the classification of the basic movement types using RNN for 5-s window are shown in Table 3.

The only result standing out in the confusion matrix in Figure 9b is the communication category. However, the “low” accuracy is explained by checking at the distribution which consisting of the fewest sliding windows for communication which is expected to be the hardest class to predict for the RNN.

**Ten seconds sliding window:** The highest performing combination of hyper-parameters for ten second sliding window is also the highest performing result for basic movements in general. Combining the Adam optimizer with a learning rate of 0.001, in a RNN with four cells of size 32, an overall accuracy of 98.75% and an overall F1-score of 98.06% is achieved (see Table 5. The recall (R), precision (P), and F1-score (F1) percentages about the classification of the basic movement types using RNN for 10-s window are shown in Table 3. Considering the overlapping of data-point in the sliding windows, and the fact that RNN uses prior knowledge to improve, these results are expected. The few miss-interpreted sliding windows are assumed to be caused by the “noise” from the placement of the accelerometer. Figure 9c shows the confusion matrix of the best result for basic movement of 10-s window using RNN.

#### 4.2.2. Experiments with Wrist-Worn Dataset—Specific Movement

Throughout this section, we explain and discuss the results of the best performing hyper-parameters for all three types of sliding windows for specific movement in the wrist-worn dataset.

**Three seconds sliding window:** The results of the RNN for three second sliding window are noticeably better than the DNN results. A RNN with four cells, with size 32, a learning rate of 0.001 and the Adam optimizer, gives an overall accuracy of 70.39% and an overall F1-score of 65.58% (see Table 5). One of the main differences between the results of the RNN and the DNN are both descending and climbing the stairs. The recall (R), precision (P) and F1-score (F1) percentages about the classification of the specific movement types using RNN for 3-s window are shown in Table 3. Figure 10a shows that the RNN has reduced the number of miss-interpretations of stair movements as walking. Thus, we assume that the “memory” of the RNN are able to remember the small differences between walking and moving upwards or downwards.

Many of the miss-interpreted movements are movements of similar types, mostly different hand movements. Thus, some sliding windows are interpreted incorrectly as another hand-movement. Examples of such miss-interpretations are drinking which may be interpreted as pouring water, Eating with a spoon which may be interpreted as pouring water. These movements are all hand-gestures which understandably can be confused with each other, and not a mis-classification of high relevance since it is of little importance to physical activity epidemiology.

The bad results of body movements from lying down, sitting down, and standing up has a low precision score due to their low data distribution. Considering the low amount of sliding windows, these classes are misinterpreted as a few different categories. However, they are mostly interpreted as similar movements. Lying down as either getting out of bed, sitting down or walking, sitting down as standing up, and standing up as getting out of bed.

**Five seconds sliding window:** An overall accuracy of 85.6% and an overall F1-score of 81.67% is the highest performing result of five second sliding window with RNN (see Table 5). The RNN used to accomplish these results is a network consisting on 4 cells of size 32. This network uses Adam as its optimizer and a learning rate of 0.01.

The results of the five second sliding window are overall increased compared to the three second sliding window. This is probably due to the overlap in the sliding windows, which further allows the network to recognize the differences in the movement patterns. Again, the worst performing categories are the ones with the lowest distribution of sliding windows. Thus, they are interpreted as movements with similar patterns to its own. The recall (R), precision (P), and F1-score (F1) percentages about the classification of the specific movement types using RNN for 5-s window are shown in Table 3. Figure 10b shows the confusion matrix of the best result for specific movement five second window using RNN.

**Ten second sliding window:** Ten second sliding windows are again the highest performing distribution of the data-points. Achieving an overall accuracy of 96.52% and an overall F1-score of 93.43%, when using Adam as the optimizer, a learning rate of 0.001 on a RNN consisting of 8 cells of size 32 and this is seen in Table 5. The recall (R), precision (P), and F1-score (F1) percentages about the classification of the specific movement types using RNN for 10-s window are shown in Table 3.

Figure 10c shows the confusion matrix of the best result for specific movement 10-s window using RNN. Most miss-interpretations are eliminated with the exception of lying down, the category with the fewest sliding windows. There are some concerns to these results as the small shift of one second for each ten second sliding window might cause the algorithms to over-fit during training. Thus, achieving such impressive results. The problem however is the size of the dataset, increasing the shift of the sliding window drastically reduces the number of sliding windows in the dataset. Leading to poor training as the size of the dataset would be low.

#### 4.2.3. Experiments with Hip-Worn Dataset

In this section, we discuss the highest performing results for the RNN on the hip-worn dataset which is collected ourselves from voluntary participants. This dataset is also tested for each type of sliding windows, which are separately discussed throughout this section. The hip-worn dataset is tested using only the RNN model, with the different hyper-parameter combinations discussed above. The decision to not to run this dataset through the DNN model is the fact that the results of the wrist-worn dataset shows that the RNN model consistently outperforms the DNN for this type of time series movement patterns.

**Three second sliding window:** The highest performing result for the three second sliding window gets an overall accuracy of 85.5%, and a F1-score of 84.04% (see Table 5). Examining the results shown in the confusion matrix in Figure 11a, the wrongly interpreted sliding windows are reasonable. Examples are laying down which is interpreted as sitting relaxed, which in some cases might be a person almost lying in a sofa. Walking is sometimes confused with walking fast, which might be explained by differences in walking speed between participants. One person’s normal walking speed might be the same speed as another person’s speed when walking fast. Thus, may be confusing the algorithm.

Looking at the categories which are shown in Table 4, and their individual performances which are shown in Figure 11a, the lower performing category is sitting in a vehicle. As there are multiple vehicle options, it can be hard to predict this category. For instance, if one of the participants where sitting in a bus, then the three second sliding window has a possibility to be when the bus is at a stop. Thus, making the algorithm believe it is a person who is sitting still. The results explained above are achieved through a RNN with 4 cells, with cell sizes of 32, a learning rate of 0.1 and Adagrad as its optimizer. The recall (R), precision (P), and F1-score (F1) percentages about the classification of the specific movement types using RNN for 10-s window are shown in Table 3.

**Five second sliding window:** Using Adagrad as the optimizer for a RNN with four cells, of size 32, combined with a 0.1 learning rate, an accuracy of 88.48% and a F1-score of 85.29% is obtained (see Table 5). Again the category of sitting in a vehicle is among the lowest performing categories, as different vehicles have different driving patterns which might confuse it with other categories, and vibration from the vehicle and ground may introduce noise and since the logs are self-reported, we cannot rule out that some reporting is unreliable. However, looking at the results of both the category for walking stair and walking fast, shown in the confusion matrix in Figure 11b, their recall score decreased compared to the three second sliding window.

The recall (R), precision (P) and F1-score (F1) percentages about the classification of the hip-worn data using RNN for 5-s window are shown in Table 4. We assume that the patterns between walking stairs and walking in normal speed are easier to distinguish when using three seconds of data compared to five. Thus, the results gets worse for five second sliding window within this category.

**Ten second sliding window:** A clear pattern in the results is that ten second sliding windows perform better compared to the three- and five- second sliding windows. Testing the RNN with the hip-worn dataset is no exception. When combining a learning rate of 0.01 with a RNN with four cells, with a size of 32, and using Adam as the optimizer, an accuracy of 89.31% with a F1-score of 89.36% is achieved (see Table 5).

The results shown in the confusion matrix in Figure 11c are impressively high. Most categories are interpreted correctly more than 85% of the time with the exception of sitting in a vehicle and walking fast. Other incorrectly interpreted sliding windows are confused with related categories, such as standing interpreted as sitting, and sitting relaxed as laying still. The recall (R), precision (P), and F1-score (F1) percentages about the classification of the hip-worn data movement types using RNN for 10-s window are shown in Table 4.

## 5. Discussion

This paper has used and presented two deep learning models for the classification of physical movement patterns from on-body accelerometer sensors. Our models were trained on two on-body accelerometer sensor datasets.

For recognizing ADLs with DNNs, few categories are accomplished with good success. The DNN achieve accuracies between 80–93% and F1-scores between 80–90% for basic movements (see Table 6). When increasing the complexity of the dataset by using specific movement categories, the accuracies significantly decrease. This is expected as there are more categories to learn and recognize. The achieved accuracy for the specific movement types, using DNN, is between 60–80%, while the F1-scores are between 55–65% (see Table 6). Considering each experiment trained the model for 2000 training steps, then providing it with unseen data to classify, the DNN is able to classify “unknown” data.

When different combinations of hyper-parameters are used, the performance is depending on which categorization of the movement patterns was used. For the broad categories, ADLs, low learning rates, and the Adam optimizer achieve the highest F1 scores, i.e., 78.5% for three seconds, 82.4% for five seconds, and 89.4% for ten second sliding windows. Recognizing the specific movements, Adagrad, with a learning rate of 0.01 gets the highest results, i.e., 56.6% for three seconds, 62.1% for five seconds, and 67.8% for ten second sliding windows.

If we compare whether the DNN model performs better than the state-of-the-art algorithms, the state-of-the-art algorithms perform better at some categories such as drinking from a glass, climbing stairs, pouring water into a glass, and standing up from a chair. In contrast, the DNN model performs better at the others getting out of bed, sitting down on a chair, and walking. Additionally, the DNN model is trained to recognize all of the categories in the wrist-worn dataset, not just a selection.

Table 6 shows results of both DNN and RNN for all 14 specific movements of wrist-worn data. Analyzing Table 6, the DNN model achieves relatively good results considering the complexity of the classification. Comparing it to [27], which classifies seven categories, the DNN achieves comparable results using all 14 categories. Thus, we argue that the DNN model, at the very least, matches the performance of the algorithm used in [27].

When comparing our RNN model to our DNN model, the accuracies are consistently higher for the used RNN model. They are achieving accuracies between 85–99% and F1-scores between 83–98% for the basic movement types. For the specific movement, the accuracies are between 70–97%, while the F1-scores are between 65–94% (see Table 6).

In addition to the ADLs and specific movement types for the wrist-worn dataset, we also tested the hip-worn dataset with the RNN. The results of this dataset are impressive and the most important finding in this study from an epidemiological perspective, considering that it consists of movements of similar type, with different intensities in a free-living setting. They are achieving accuracies between 85–90% and F1-scores of 84–90%. Thus, showing that the RNN model is able to perform at a high level on new datasets.

Further, when we compare our results with the existing literature (see Table 7), for classifying the physical movement activities, our results show higher accuracy when RNN is used, but compared with all other literature, they have a larger and different dataset, making the results incomparable.

## 6. Conclusions and Future Work

This paper presents two distinct deep learning approaches, i.e., DNN and RNN, to classify physical movement patterns from body-worn tri-axial accelerometer data. We performed numerous experiments with different combinations of hyper-parameters, optimizers, and learning rates for the classification of the movement patterns. Both the models were trained on two separate accelerometer datasets. The first dataset, from the UCI machine learning repository, contains 14 various activities of daily life (ADL) collected from 16 volunteers who carried a single wrist-worn tri-axial accelerometer. The second dataset, collected by us, has ten different ADLs from eight volunteers who placed the sensors on their hip and carried them out during their daily activities.

Through experiments, both the DNN and RNN models were evaluated under several performance metrics such as precision, recall, F1-score, and accuracy. Our experimental results showed that the DNN model is able to recognize “unlabeled” data with an acceptable overall recall percentage of 64%, using 14 categories, which is a 10% increase compared to the state-of-the-art algorithms, which has only a 54% overall recall score for seven types. Whereas the RNN model performs significantly better on the time-series data, reaching an overall recall score of 94%, which is a 40% increase compared to the DNN. The RNN model results surpass the percentages for most categories compared to the state-of-the-art, even when classifying all classes in the dataset. Furthermore, the RNN model is able to recognize different movement patterns from a new dataset consisting of movement types of varying intensities. Thus, our results show that the RNN model is the best-suited algorithm of the discussed algorithms for movement pattern recognition.

As future work, we intend to extend this work to experiment with restructuring the deep learning models first to recognize these broad ADL categories, then classify which specific movement type within the ADL it is and test different optimizers in combination with varying rates of learning and also experiment with different algorithms such as Convolutional Neural Networks and Support Vector Machines.

## Figures and Tables

**Figure 1 sensors-21-05564-f001:**
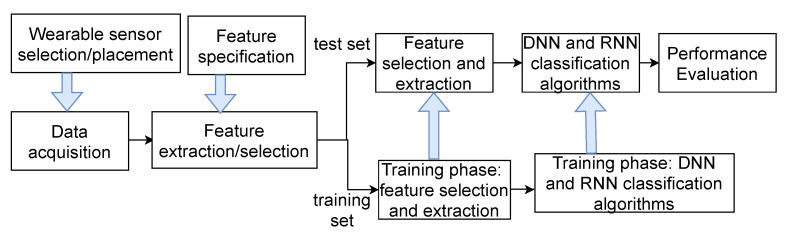
Research Methodology.

**Figure 2 sensors-21-05564-f002:**
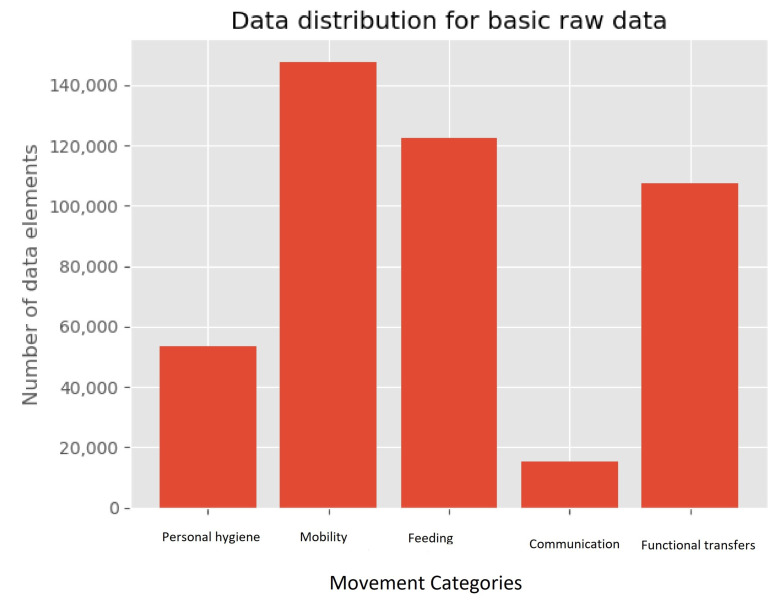
Basic Raw data distribution for the wrist-worn dataset.

**Figure 3 sensors-21-05564-f003:**
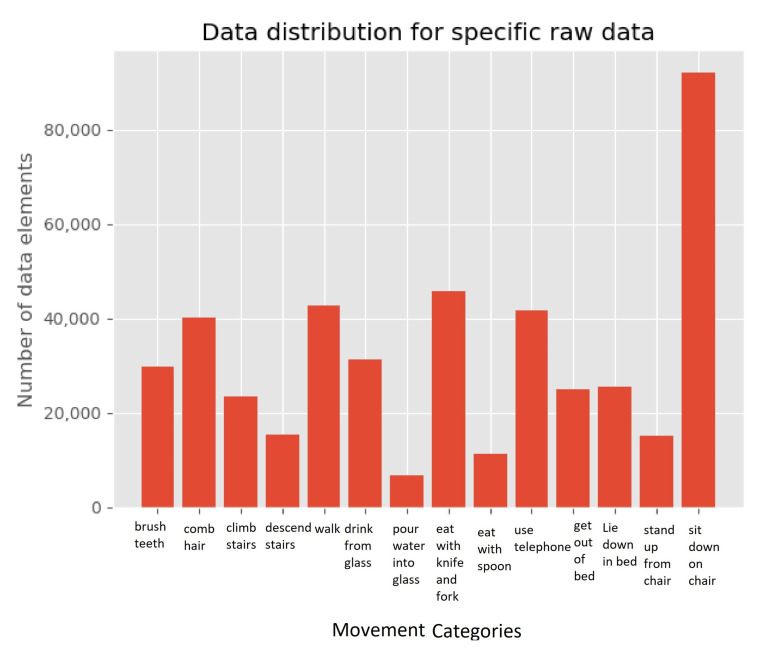
Specific raw data distribution for the wrist-worn dataset.

**Figure 4 sensors-21-05564-f004:**
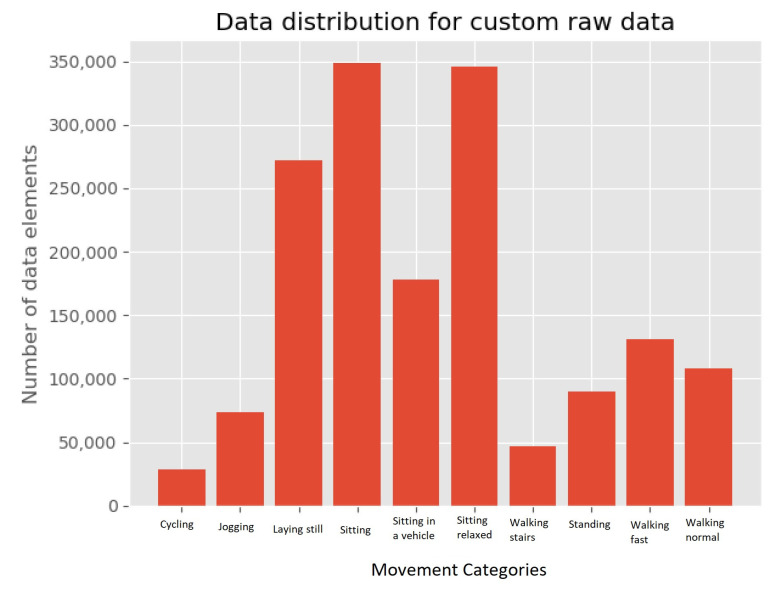
Raw data distribution for the hip-worn dataset.

**Figure 5 sensors-21-05564-f005:**
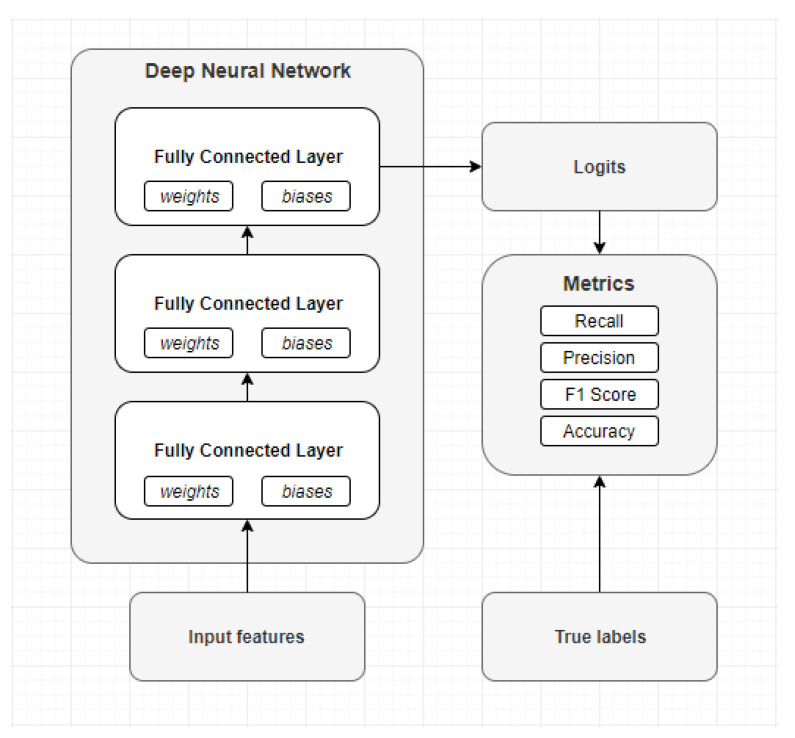
Structure of the Deep Neural Network model.

**Figure 6 sensors-21-05564-f006:**
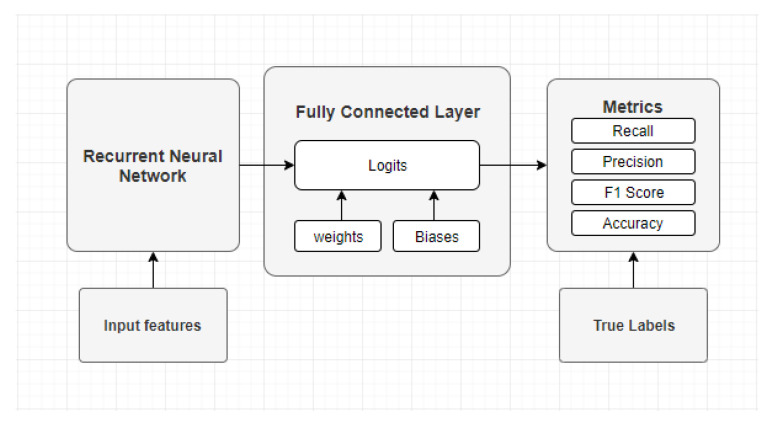
Structure of the Recurrent Neural Network model.

**Figure 7 sensors-21-05564-f007:**
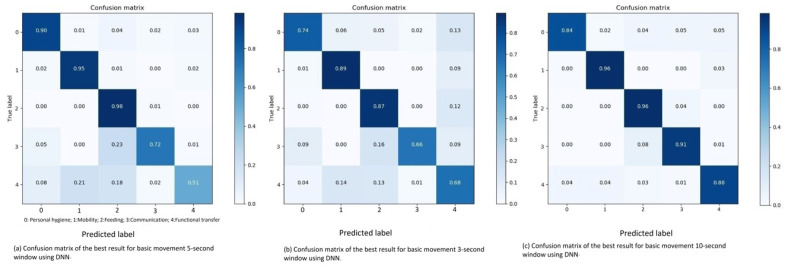
Confusion matrix of the best result for basic movement 3, 5 and 10-s window using DNN.

**Figure 8 sensors-21-05564-f008:**
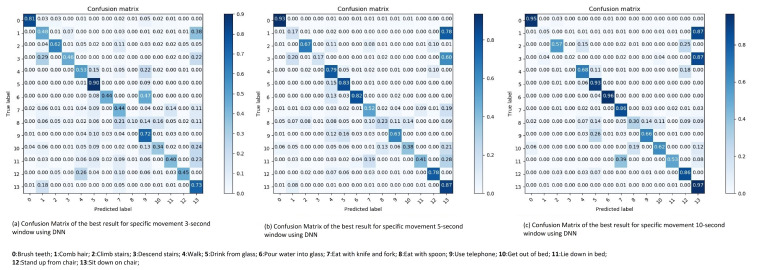
Confusion matrix of the best result for specific movement 3, 5, 10-s window using DNN.

**Figure 9 sensors-21-05564-f009:**
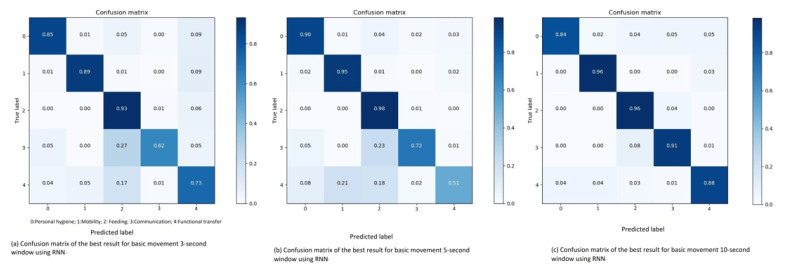
Confusion matrix of the best result for basic movement 3, 5, 10-s window using RNN.

**Figure 10 sensors-21-05564-f010:**
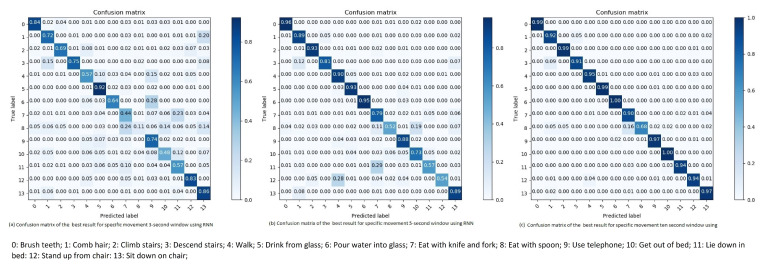
Confusion matrix of the best result for specific movement 3, 5 and 10-s window using RNN.

**Figure 11 sensors-21-05564-f011:**
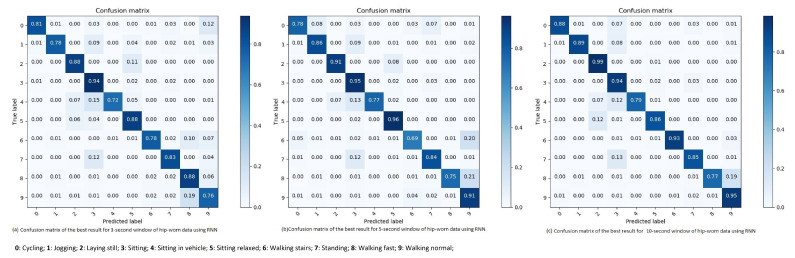
Confusion matrix of the best result for 3, 5, 10-s window of hip-worn data using RNN.

**Table 1 sensors-21-05564-t001:** Correlation between ADL categories and Motion primitives.

ADL	Motion Primitives
Personal hygiene	Brush teeth
	Comb Hair
Mobility	Climb Stairs
	Descend Stairs
	Walk
Feeding	Drink from glass
	Pour water into glass
	Eat with knife and fork
	Eat with spoon
Communication	Use telephone
Functional transfers	Get out of bed
	Lie down in bed
	Stand up from chair
	Sit down on chair

**Table 2 sensors-21-05564-t002:** Best result of basic and specific movement of wrist-worn dataset 3, 5, 10-s window using DNN.

Category	Basic Movement with DNN								
	3-W ^1^			5-W ^2^			10-W ^3^		
	R ^a^	P ^b^	F_1_ ^c^	R ^a^	P ^b^	F_1_ ^c^	R ^a^	P ^b^	F_1_ ^c^
Hygiene	73.7%	83.61%	78.35%	89.69%	84.33%	86.93%	84.1%	96.49%	89.87%
Mobility	89.33%	88.7%	89.01%	94.53%	89.38%	91.88%	95.95%	97.51%	96.72%
Feeding	87.13%	84.44%	85.76%	98.37%	83.69%	90.43%	95.87%	93.67%	94.75%
Communication	66.23%	79.69%	72.34%	71.72%	72.82%	72.26%	91.3%	70.59%	79.62%
F-Transfer	67.57%	64.6%	66.05%	50.54%	88.89%	64.44%	87.74%	81.4%	84.45%
	**Specific Movement with DNN**								
	**3-W ^1^**			**5-W ^2^**			**10-W ^3^**		
	**R ^a^**	**P ^b^**	**F_1_ ^c^**	**R ^a^**	**P ^b^**	**F_1_ ^c^**	**R ^a^**	**P ^b^**	**F_1_ ^c^**
Brush teeth	81.29%	90.0%	85.42%	93.49%	91.56%	92.52%	94.77%	98.82%	96.75%
Climb stairs	48.07%	39.55%	43.39%	16.95%	32.72%	22.33%	4.9%	35.0%	8.59%
Comb hair	61.82%	79.07%	69.39%	17.29%	79.31%	28.4%	57.26%	84.81%	68.37%
Descend stairs	46.15%	57.69%	51.28%	17.29%	79.31%	28.4%	2.17%	33.33%	4.08%
Drink	52.74%	64.63%	58.08%	79.06%	59.97%	68.2%	68.28%	60.77%	64.3%
Eat w/knife&fork	90.0%	60.71%	72.51%	83.3%	74.58%	78.7%	93.01%	82.06%	87.19%
Eat w/ spoon	44.44%	59.26%	50.79%	44.44%	59.26%	50.79%	95.79%	93.81%	94.79%
Get out bed	43.75%	49.46%	46.43%	51.93%	68.18%	58.96%	85.66%	83.27%	84.44%
Lie down bed	9.52%	66.67%	16.67%	23.42%	37.68%	28.89%	29.55%	61.9%	40.0%
Pour water	72.14%	52.73%	60.92%	63.38%	80.12%	70.77%	66.48%	85.21%	74.69%
Sit down	34.12%	46.03%	39.19%	38.06%	68.6%	48.96%	62.5%	60.61%	61.54%
Stand up	39.58%	39.58%	39.58%	40.67%	49.19%	44.53%	52.78%	55.88%	54.29%
Telephone	44.87%	77.78%	56.91%	78.26%	66.12%	71.68%	85.54%	58.44%	69.44%
Walk	72.75%	67.41%	69.98%	87.47%	64.0%	73.91%	96.55%	83.3%	89.44%

^1^ Three-second window, ^2^ Five-second window, ^3^ Ten-second window, ^a^ Recall, ^b^ Precision, ^c^ F1-score.

**Table 3 sensors-21-05564-t003:** Best result of basic and specific movement of wrist-worn dataset 3, 5, 10-s window using RNN.

Category	Basic Movement with RNN								
	3-W ^1^			5-W ^2^			10-W ^3^		
	R ^a^	P ^b^	F_1_ ^c^	R ^a^	P ^b^	F_1_ ^c^	R ^a^	P ^b^	F_1_ ^c^
Hygiene	84.81%	88.42%	86.58%	96.95%	93.73%	95.31%	99.85%	98.79%	99.32%
Mobility	88.62%	95.61%	91.98%	94.86%	98.36%	96.58%	99.58%	99.08%	99.33%
Feeding	93.12%	80.86%	86.56%	97.16%	95.66%	96.41%	99.66%	98.35%	99.0%
Communication	62.34%	85.71%	72.18%	83.33%	91.67%	87.3%	92.93%	99.42%	96.07%
F-Transfer	73.22%	73.53%	73.38%	90.85%	87.79%	89.29%	94.71%	98.27%	96.45%
	**Specific Movement with RNN**								
	**3-W ^1^**			**5-W ^2^**			**10-W ^3^**		
	**R ^a^**	**P ^b^**	**F_1_ ^c^**	**R ^a^**	**P ^b^**	**F_1_ ^c^**	**R ^a^**	**P ^b^**	**F_1_ ^c^**
Brush teeth	83.87%	86.09%	84.97%	96.01%	96.21%	96.11%	99.32%	99.77%	99.54%
Climb stairs	72.38%	71.2%	71.78%	88.78%	75.15%	81.4%	92.31%	85.71%	88.89%
Comb hair	69.09%	83.52%	75.62%	92.54%	92.23%	92.39%	98.72%	98.72%	98.72%
Descend stairs	75.38%	87.5%	80.99%	81.2%	81.82%	81.51%	91.3%	84.0%	87.5%
Drink	56.72%	73.08%	63.87%	90.42%	81.36%	85.65%	95.16%	94.65%	94.91%
Eat w/knife&fork	92.35%	73.02%	81.56%	92.9%	87.77%	90.26%	98.94%	99.36%	99.15%
Eat w/ spoon	63.89%	50.0%	56.1%	94.68%	89.9%	92.23%	100%	95.0%	97.44%
Get out bed	44.23%	57.86%	50.14%	79.23%	77.34%	78.27%	90.16%	94.02%	92.05%
Lie down bed	11.11%	36.84%	17.07%	52.25%	63.04%	57.14%	68.18%	85.71%	75.95%
Pour water	74.13%	65.35%	69.46%	88.03%	84.84%	86.41%	97.25%	95.68%	96.46%
Sit down	48.24%	45.56%	46.86%	72.9%	68.48%	70.62%	100%	88.89%	94.12%
Stand up	57.29%	37.67%	45.45%	57.33%	69.92%	63.0%	94.44%	85.0%	89.47%
Telephone	83.33%	66.33%	73.86%	54.11%	95.73%	69.14%	93.98%	93.98%	93.98%
Walk	86.15%	84.3%	85.22%	89.14%	93.84%	91.43%	97.49%	98.42%	97.95%

^1^ Three-second window, ^2^ Five-second window, ^3^ Ten-second window, ^a^ Recall, ^b^ Precision, ^c^ F1-score.

**Table 4 sensors-21-05564-t004:** Best result of 3, 5, 10-s window of hip-worn data using RNN.

Category	Movements in Hip-Worn Data with RNN								
	3-W ^1^			5-W ^2^			10-W ^3^		
	R ^a^	P ^b^	F_1_ ^c^	R ^a^	P ^b^	F_1_ ^c^	R ^a^	P ^b^	F_1_ ^c^
Cycling	81.33%	82.99%	82.15%	78.08%	78.74%	78.41%	87.55%	89.67%	88.6%
Jogging	77.67%	94.28%	85.17%	85.82%	89.39%	87.57%	89.41%	96.54%	92.84%
Laying still	87.8%	86.82%	87.31%	90.61%	94.86%	92.69%	98.52%	82.88%	90.02%
Sitting	93.73%	83.56%	88.36%	94.84%	86.27%	90.35%	94.27%	87.44%	90.73%
Sitting in a vehicle	72.09%	98.53%	83.26%	76.81%	97.51%	85.93%	79.01%	98.36%	87.63%
Sitting relaxed	88.19%	85.55%	86.85%	96.19%	90.63%	93.33%	85.97%	96.12%	90.76%
Walking stairs	77.96%	83.04%	80.42%	68.61%	78.83%	73.37%	93.18%	94.41%	93.79%
Standing	83.27%	82.45%	82.85%	84.24%	80.89%	82.53%	85.09%	81.83%	83.43%
Walking fast	87.97%	81.27%	84.49%	74.51%	98.56%	84.86%	77.3%	98.55%	86.64%
Walking normal	75.63%	77.2%	76.41%	90.78%	70.02%	79.06%	94.69%	76.51%	84.63%

^1^ Three-second window, ^2^ Five-second window, ^3^ Ten-second window, ^a^ Recall, ^b^ Precision, ^c^ F1-score.

**Table 5 sensors-21-05564-t005:** Summary of the DNN and RNN results.

DNN			
**Dataset**	**Sliding Window**	**Accuracy**	**F1 Score**
Wrist Worn: Basic Movement	3 s	80.94%	78.46%
	5 s	86.01%	82.37%
	10 s	**92.4%**	**89.43%**
Wrist-Worn: Specific Movement	3 s	59.93%	56.59%
	5 s	67.83%	62.11%
	10 s	**81.29%**	**66.75%**
**RNN**			
Wrist Worn: Basic Movement	3 s	84.89%	82.56%
	5 s	94.65%	93.03%
	10 s	**98.75%**	**98.06%**
Wrist-Worn: Specific Movement	3 s	70.39%	65.58%
	5 s	85.6%	81.67%
	10 s	**96.52%**	**93.43%**
Hip-Worn	3 s	85.5%	84.04%
	5 s	88.48%	85.29%
	10 s	**89.31%**	**89.36%**

**Table 6 sensors-21-05564-t006:** Results of both DNN and RNN for specific movements of wrist-worn data.

	cDNN Results	RNN Results				
**Category**	**Recall**	**Precision**	**F1 Score**	**Recall**	**Precision**	**F1 Score**
Brush teeth	94.77%	98.82%	96.75%	99.32%	99.77%	99.54%
Comb hair	57.26%	84.81%	68.37%	98.72%	98.72%	98.72%
Climb stairs	4.9%	35.0%	8.59%	92.31%	85.71%	88.89%
Walk	96.55%	83.3%	89.44%	97.49%	98.42%	97.95%
Descend stairs	2.17%	33.33%	4.08%	91.3%	84.0%	87.5%
Drink from glass	68.28%	60.77%	64.3%	95.16%	94.65%	94.91%
Pour water into glass	66.48%	85.21%	74.69%	97.25%	95.68%	96.46%
Eat w/ fork and knife	93.01%	82.06%	87.19%	98.94%	99.36%	99.15%
Eat w/spoon	95.79%	93.81%	94.79%	100%	95.0%	97.44%
Use telephone	85.54%	58.44%	69.44%	93.98%	93.98%	93.98%
Get out of bed	85.66%	83.27%	84.44%	90.16%	94.02%	92.05%
Lie down in the bed	29.55%	61.9%	40.0%	68.18%	85.71%	75.95%
Stand up from chair	52.78%	55.88%	54.29%	94.44%	85.0%	89.47%
Sit down on chair	62.5%	60.61%	61.54%	100%	88.89%	94.12%

**Table 7 sensors-21-05564-t007:** Results ofboth DNN and RNN for specific movements of wrist-worn data.

LR	Dataset Type	Method	Accuracy
[23]	UCI HAR	iSPLInception	95%
[24]	UCI HAR	LSTM	52.77%
[25]	UCI HAR	LSTM	96.44%
		MFAP	98.85%
[27]	UCI HAR	GMM	-
		GMR	-
[28]	UCI HAR	LSTM	97.4%
[26]	UCI HAR	1D CNN	65%
		2D CNN	76%
		2-layer LSTM	85%
		3-layer LSTM	91%
		Bi-directional LSTM	93%
		LSTM	77%
		1D CNN+LSTM	76%
[29]	UCI HAR	Uni-directional	96.7%
		DRNN	95.2%
		CNN	96%
		SVM	93.3%
[30]	UCI HAR	DBN	97.5%
		SVM	94.12%
[32]	Wrist-worn	Naive Bayesian	97.4%
		SVM	97.8%
		C4.5	93%
		GMM	92.2%
		Logistic classifier	94.0%
		kNN	98.5%
		Parzen classifier	92.7%
		ANN	93%
[31]	Wrist-worn	Logistic regression	88.4% ± 3.0%
	Hip-worn		91.0% ± 3.1%
[34]	Wrist-worn	CNN-SVM	0.850
	Thigh	CNN-kNN	0.845
[33]	Hip-worn	GMM	91.3%
		Heuristic	71%
**Our work**	**UCI- Basic**	**DNN**	**92.4%**
	**UCI- Specific**	**DNN**	**81.29%**
	**UCI- Basic**	**RNN**	**98.75%**
	**UCI- Specific**	**RNN**	**96.52%**
	**Hip-worn**	**RNN**	**89.31%**

## Data Availability

Data collected through research presented in the paper are available on request from the corresponding authors.

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
