# Peer review of "Deep Learning for Classifying Physical Activities from Accelerometer Data"

_sensors, 2021, doi:10.3390/s21165564_

Round 1

Reviewer 1 Report

Overall the paper is well written and interesting.

Albeit the innovative content is fairly limited, the study is relevant and the sensitivity analysis conducted on the window size is an interesting read.

In the state of the art section the authors give great focus on the performance reached by the various works, that in my opinion are not relevant: they were obtained on different datasets and in different tasks, so they are in most cases just number that take out from the discussion of the different purposes and goals their systems implemented.

In this direction I believe you could discuss briefly also some works from the very similar research topic of transportation mode recognition with deep neural networks. A lot of works use the very same smartphone data as yours and the reader may be interested in their NN architectures.

The authors should also strengthen the innovation points of their work in section 1, as it is quite lacking and the greatest shortcoming of the work.

Consider adding a convolutional neural network to the test, as it can deal with this kind of time series and SVMs as representatives for classic ML approaches. Consider also open-sourcing your code to maximise impact.

Several figures with the confusion matrices are not readable.

Author Response

Overall the paper is well written and interesting.Albeit the innovative content is fairly limited, the study is relevant and the sensitivity analysis conducted on the window size is an interesting readIn the state of the art section the authors give great focus on the performance reached by the various works, that in my opinion are not relevant: they were obtained on different datasets and in different tasks, so they are in most cases just number that take out from the discussion of the different purposes and goals their systems implemented.

In this direction I believe you could discuss briefly also some works from the very similar research topic of transportation mode recognition with deep neural networks. A lot of works use the very same smartphone data as yours and the reader may be interested in their NN architectures.

Updated section 2.

The authors should also strengthen the innovation points of their work in section 1, as it is quite lacking and the greatest shortcoming of the work.

Revised the section 1 and added our contribution in line numbers 37-58.

Consider adding a convolutional neural network to the test, as it can deal with this kind of time series and SVMs as representatives for classic ML approaches. Consider also open-sourcing your code to maximise impact.

It is considered as a future work and we can provide the code upon request.

Several figures with the confusion matrices are not readable.

Updated

Reviewer 2 Report

Overall, I am very happy with this submission.  I have no major objections to the work as it stands, and only minor suggestions (and they always are suggestions) to the authors.  These include:

  • I think that your review of previous work could be improved in structure.  At the moment it is merely a length enumeration/chronicling of what people have done.  A good review organizes previous work in a meaningful way, in a manner that facilitates easier understanding of what does and what does not work, where the gaps are, etc.
  • Please never write something like "In [1] ... ".  References are not a part of the text.  The above reads as "In ...".  Rather, write "Jones et al. [1] ..."

  • Please do not write "subsections 3.2.1 and 3.2.2".  The standard is to write "Sections 3.2.1 and 3.2.2" (no prefix, capitalized).

  • There is a space missing in Line 369 in "The recall(R)".  The same is the case in a few other places.

Author Response

Please find our replies to the comments given by reviewer 2.

Reviewer 3 Report

The authors have presented an interesting work about physical activities classification using deep learning.

Although the paper is well written and presented, there are some major issues to take into account.

The main idea of the paper is to help experts to track physical activities from a long range of groups; from older people to weightlifting. This is a very important and useful goal.

However, the methods and justification were not properly presented. For instance, what is the gain of a precision of 1 or 5% in a given activity. What the gain if you run 10.000 or 10.050 meters? Why do you need such precision considering the goal is a general monitoring and not a high performance system?

Moreover, for that purpose, the variables in the training dataset are also not appropriate. It should include more well defined activities rather than general purpose ones.

The literature review is also simple. There are a lot of similar works  that deals with such problem with very good results.

All in all, you should have:

 a better literature review to test and compare with state-of-the-art algorithms,  

to improve the justification. It must be clearly defined why to have such precision is important

to define what you are looking for in terms of gain. You defined that is to monitor physical activities, what for exactly? What are the best inputs to define this problem? A general UCI dataset is enough? Or the paper is just about classification without any real propose? If so, several other datasets should be tested and compared.

Author Response

Please find our replies to the comments given by reviewer 3.

Reviewer 4 Report

Dear Authors, 

Please find the attached file for my comments. 

Best Regards 

Author Response

Please find our replies to the comments given by reviewer 4.

Round 2

Reviewer 1 Report

I have no other comments

Author Response

Thank you.

Reviewer 3 Report

the paper can be accepted for publication.

Author Response

We have gone through the paper once again and made the necessary changes in terms of the English language.

Reviewer 4 Report

Dear Authors, 

Thank you for addressing my comments and I have some minor comments. Please check the paper based on the minor comments.

  1. Please check the affiliation details.
  2. In the sentence of the abstract, the word ‘our’ is repeated, please check the sentence.
  3. In the last paragraph of the introduction, please add the rest of the paper is organized as follows.
  4. 1 change to Figure 1.
  5. Please add a full stop (.) at the end of the figure and table captions.
  6. In Figures 5 and 7, same as the x-axis, please change the numbers into label names in Y-axis.

Best Regards.
Reviewer 

Author Response

Hi,

Thank you for reviewing our paper. We have made changes according to the suggestions. Please see below.

  1. Please check the affiliation details. Done
  2. In the sentence of the abstract, the word ‘our’ is repeated, please check the sentence. Done. See line 14 in the pdf.
  3. In the last paragraph of the introduction, please add the rest of the paper is organized as follows. Done. See line 57 in the pdf.
  4. 1 change to Figure 1. Done. See line 176 in the pdf.
  5. Please add a full stop (.) at the end of the figure and table captions. Done
  6. In Figures 5 and 7, same as the x-axis, please change the numbers into label names in Y-axis. Done